# Neural Networks Fail to Learn Periodic Functions and How to Fix It

**Liu Ziyin**[1]**, Tilman Hartwig**[1,2,3]**, Masahito Ueda**[1,2,4]
[1]Department of Physics, School of Science, The University of Tokyo
[2]Institute for Physics of Intelligence, School of Science, The University of Tokyo
[3]Kavli IPMU (WPI), UTIAS, The University of Tokyo
[4]RIKEN CEMS

## Abstract

Previous literature offers limited clues on how to learn a periodic function using modern neural networks. We start with a study of the extrapolation properties of neural networks; we prove and demonstrate experimentally that the standard activations functions, such as ReLU, tanh, sigmoid, along with their variants, all fail to learn to extrapolate simple periodic functions. We hypothesize that this is due to their lack of a "periodic" inductive bias. As a fix of this problem, we propose a new activation, namely, $x + \sin^2(x)$, which achieves the desired periodic inductive bias to learn a periodic function while maintaining a favorable optimization property of the $\mathrm{ReLU}$-based activations. Experimentally, we apply the proposed method to temperature and financial data prediction.

## 1  Introduction

In general, periodic functions are one of the most basic functions of importance to human society and natural science: the world's daily and yearly cycles are dictated by periodic motions in the Solar System [26]; the human body has an intrinsic biological clock that is periodic in nature [20, 35], the number of passengers on the metro follows daily and weekly modulations, and the stock market experiences (semi-)periodic fluctuations [28, 43]. Global economy also follows complicated and superimposed cycles of different periods, including but not limited to the Kitchin and Juglar cycle [10, 22]. In many scientific scenarios, we want to model a periodic system in order to be able to predict the future evolution, based on current and past observations. While deep neural networks are excellent tools in interpolating between existing data, their fiducial version is not suited to extrapolate beyond the training range, especially not for periodic functions.

If we know beforehand that the problem is periodic, we can easily solve it, e.g., in Fourier space, or after an appropriate transformation. However, in many situations we do not know a priori if the problem is periodic or contains a periodic component. In such cases it is important to have a model that is flexible enough to model both periodic and non-periodic functions, in order to overcome the bias of choosing a certain modelling approach. In fact, despite the importance of being able to model periodic functions, no satisfactory neural network-based method seems to solve this problem. Some previous methods that propose to use periodic activation functions exist [38, 45, 30]. This line of works propose using standard periodic functions such as $\sin(x)$ and $\cos(x)$ or their linear combinations as activation functions. However, such activation functions are very hard to optimize due to large degeneracy in local minima [30], and the experimental results suggest that using $\sin$ as the activation function does not work well except for some very simple model, and that it can not compete against $\mathrm{ReLU}$-based activation functions [34, 7, 25, 42] on standard tasks.

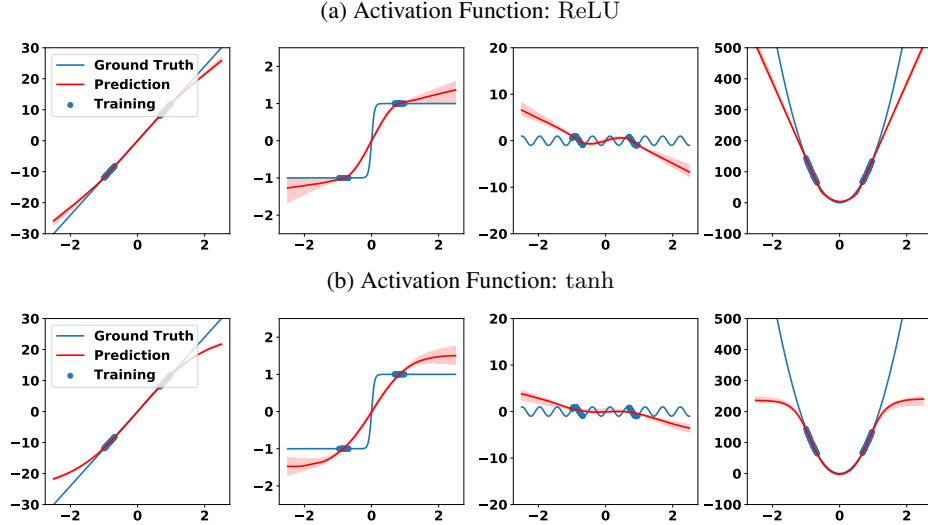

Figure 1: Exploration of how different activation functions extrapolate various basic function types: $y = x$ (first column), $y = \tanh(x)$ (second column), $y = \sin(x)$ (third column), and $y = x^2$ (last column). The red curves represents the median model prediction and the shaded regions show the 90% credibility interval from 21 independent runs. Note that the horizontal range is re-scaled so that the training data lies between $-1$ and $1$.

The contribution of this work is threefold: (1) we study the extrapolation properties of a neural network beyond a bounded region; (2) we show that standard neural networks with standard activation functions are insufficient to learn periodic functions outside the bounded region where data points are present; (3) we propose a handy solution for this problem and it is shown to work well on toy examples and real tasks. However, the question remains open as to whether better activation functions or methods can be designed.

## 2 Inductive Bias and Extrapolation Properties of Activation Functions

A key property of periodic functions that differentiates them from regular functions is the extrapolation property of such functions. With a period $2\pi$, a period function $f(x) = f(x + 2\pi)$ repeats itself *ad infinitum*. Learning a periodic function, therefore, not only requires fitting of pattern on a bounded region, but the learned pattern needs to extrapolate beyond the bounded region. In this section, we experiment with the inductive bias that the common activation functions offer. While it is hard to investigate the effect of using different activation functions in a general setting, one can still hypothesize that the properties of the activation functions are carried over to the property of the neural networks. For example, a tanh network will be smooth and extrapolates to a constant function, while ReLU is piecewise-linear and extrapolates in a linear way.

### 2.1 Extrapolation Experiments

We set up a small experiment in the following way: we use a fully connected neural network with one hidden layer consisting of $512$ neurons. We generate training data by sampling from four different analytical functions in the interval [-5,5] with a gap in the range [-1,1]. This allows us to study the inter-and-extrapolation behaviour of various activation functions. The results can be seen in Fig. 1. This experimental observation, in fact, can be proved theoretically in a more general form. We see that their extrapolation behaviour is dictated by the analytical form of the activation function: ReLU diverges to $\pm\infty$, and tanh levels off towards a constant value.

### 2.2 Theoretical Analysis

In this section, we study and prove the *incapability of standard activation functions to extrapolate*.

**Definition 1.** (Feedforward Neural Network.) Let $f_\sigma(x) = W_h \sigma ... \sigma W_1 x$ be a function from $\mathbb{R}^{d_1} \to \mathbb{R}^{d_{h+1}}$, where $\sigma$ is the *activation function* applied element-wise to its input vector, and $W_i \in \mathbb{R}^{d_i \times d_{i+1}}$. $f_\sigma(x)$ is called a *feedforward neural network* with activation function $\sigma$, and $d_1$ is called the input dimension, and $d_{h+1}$ is the output dimension.

Now, one can show that for arbitrary feedforward neural networks the following two *extrapolation* theorems hold.

**Theorem 1.** *Consider a feed forward network $f_{\mathrm{ReLU}}(x)$, with arbitrary but fixed depth $h$ and widths $d_1, ..., d_{h+1}$. Then*

$$\lim_{z \to \infty} \|f_{\mathrm{ReLU}}(zu) - zW_u u - b_u\|_2 = 0, \tag{1}$$

*where $z$ is a real scalar, $u$ is any unit vector of dimension $d_1$, and $W_u \in \mathbb{R}^{d_1 \times d_h}$ is a constant matrix only dependent on $u$.*

The above theorem says that any feedforward neural network with $\mathrm{ReLU}$ activation converges to a linear transformation $W_u$ in the asymptotic limit, and this extrapolated linear transformation only depends on $u$, the direction of extrapolation. See Figure 1 for illustration. Next, we prove a similar theorem for $\tanh$ activation. Naturally, a $\tanh$ network extrapolates like a constant function.

**Theorem 2.** *Consider a feed forward network $f_{\tanh}(x)$, with arbitrarily fixed depth $h$ and widths $d_1, ..., d_{h+1}$. Then*

$$\lim_{z \to \infty} \|f_{\tanh}(zu) - v_u\|_2 = 0, \tag{2}$$

*where $z$ is a real scalar, $u$ is any unit vector of dimension $d_1$, and $v_u \in \mathbb{R}^{d_{h+1}}$ is a constant vector that only depends on $u$.*

We note that these two theorems can be proved through induction, and we give their proofs in the appendix. The above two theorems show that any neural network with ReLU or the tanh activation function cannot extrapolate a periodic function. Moreover, while the specific statement of the theorem applies to $\tanh$ and ReLU, it is has quite general applicability. Since the proof is only based on the asymptotic property of activation function when $x \to \pm\infty$, one can prove the same theorem for any continuous activation function that asymptotically converges to a $\tanh$ or ReLU; for example, this would include Swish and Leaky-ReLU (and almost all the other ReLU-based variants), which converge to ReLU; one can follow the same proving procedure to prove a similar theorem for each of these activation functions.

## 3 Proposed Method: $x + sin^2(x)$

As we have seen in the previous section, the choice of the activation functions plays a crucial role in affecting the interpolation and extrapolation properties of neural networks, and such interpolation and extrapolation properties in return affect the generalization of the network equipped with such activation function.

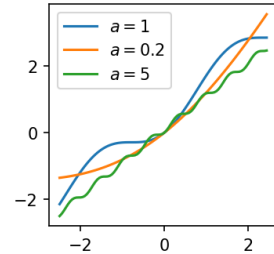

Figure 2: Snake at different $a$.

To easily address the proposed function, we propose to use $x + \sin^2(x)$ as an activation function, which we call the "Snake" function. One can augment it with a factor $a$ to control the frequency of the periodic part. Thus propose the Snake activation with frequency $a$

$$\mathrm{Snake}_a := x + \frac{1}{a}\sin^2(ax) = x - \frac{1}{2a}\cos(2ax) + \frac{1}{2a}, \tag{3}$$

We plot Snake for $a = 0.2,\ 1,\ 5$ in Figure 2. We see that larger $a$ gives higher frequency.

There are also two conceivable alternatives choices for a periodicity-biased activation function. One is the $\sin$ function, which has been proposed in [30], along with $\cos$ and their linear combinations as proposed in Fourier neural networks [45]. However, the problem of these functions does not lie in its generalization ability, but lies in its optimization. In fact, $\sin$ is not a monotonic function, and using $\sin$ as the activation function creates infinitely many local minima in the solutions (since shifting the preactivation value by $2\pi$ gives the same function), making $\sin$ hard to optimize. See Figure 3 for a comparison on training a 4-layer fully connected neural network on

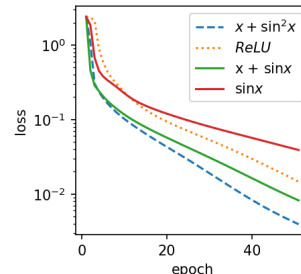

Figure 3: Optimization of different loss functions on MNIST. The proposed activation is shown as a blue dashed curve. We see that Snake is easier to optimize than other periodic baselines. Also interesting is that Snake (and $x + \sin(x)$) are also easier to train than the standard ReLU on this task.

| | ReLU | Swish | Tanh | $\sin(x)$ | $x+\sin(x)$ | $x+\sin^2(x)$ |
|---|---|---|---|---|---|---|
| monotonic | ✓ | ✗ | ✓ | ✗ | ✓ | ✓ |
| (semi-)periodic | ✗ | ✗ | ✗ | ✓ | ✓ | ✓ |
| first non-linear term | - | $\frac{x^2}{4}$ | $-\frac{x^3}{3}$ | $-\frac{x^3}{6}$ | $-\frac{x^3}{6}$ | $x^2$ |

Table 1: Comparison of different periodic and non-periodic activation functions.

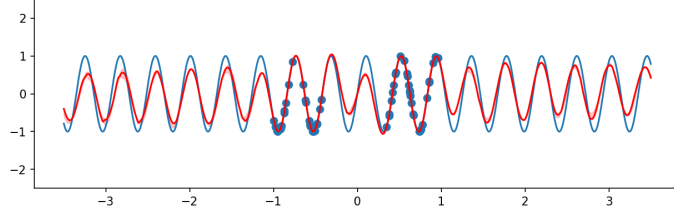

Figure 4: Regressing a simple sin function with Snake as activation functions for $a = 10$.

MNIST. We identify the root cause of the problem in $\sin$ as its non-*monotonicity*. Since the gradient of model parameters is only a local quantity, it cannot detect the global periodicity of the $\sin$ function. Therefore, the difficulty in biasing activation function towards periodicity is that it needs to achieve *monotonicity* and *periodicity* at the same time.

We also propose two other alternatives, $x + \sin(x)$ and $x + \cos(x)$. They are easier to optimize than $\sin$ similar to the commonly used $\mathrm{ReLU}$. In the neural architecture search in [34], these two functions are found to be in list of the best-performing activation functions found using reinforcement learning; while they commented that these two are interesting, no further discussion was given regarding their significance. While these two and $x+\sin^2(x)$ have the same expressive power, we choose $x+\sin^2(x)$ as the default form of Snake for the following reason. It is important to note that the preactivation values are centered around $0$ and the standard initialization schemes such as Kaiming init normalizes such preactivation values to the unit variance [40, 16]. By the law of large numbers, the preactivation roughly obeys a standard normal distribution. This makes $0$ a special point for the activation function, since most of preactivation values will lie close to $0$. However, $x + \sin(x)$ seems to be a choice inferior to $x+\sin^2(x)$ around $0$. Expanding around $0$:

$$
\begin{cases}
x + \sin(x) = 2x - \frac{x^3}{6} + \frac{x^5}{120} + o(x^5) \\
x + \sin^2(x) = x + x^2 - \frac{x^4}{3} + o(x^4).
\end{cases}
\tag{4}
$$

Of particular interest to us is the *non-linear term* in the activation, since this is the term that drives the neural network away from its linear counterpart, and learning of a non-linear network is explained by this term to leading order. One finds that the first non-linear order expansion for $x + \sin(x)$ is already third order, while that of $x+\sin^2(x)$ is contains a non-vanishing second order term, which can probe non-linear behavior that is odd in $x$. We hypothesize that this non-vanishing second order term gives Snake a better approximation property than $x+\sin(x)$. In Table 1, we compare the properties of each activation function.

*Extension:* We also suggest the extension to make the $a$ parameter a learnable parameter for each preactivation value. The benefit for this is that $a$ no-longer needs to be determined by hand. While we do not study this extension in detail, one experiment is carried out with learnable $a$, see the atmospheric temperature prediction experiment in Section 6.2.

### 3.1 Regression with Fully Connected Neural Network

In this section, we regress a simple $1$−d periodic function, $\sin(x)$, with the proposed activation function.See Figure 4 and compare with the related experiments on $\tanh$ and $\mathrm{ReLU}$ in Figure 1. As expected, all three activation functions learn to regress the training points. However, neither $\mathrm{ReLU}$ nor $\tanh$ seems to be able to capture the periodic nature of the underlying function; both baselines inter- and extrapolate in a naive way, with $\tanh$ being slightly smoother than $\mathrm{ReLU}$. On the other hand, Snake learns to both interpolate and extrapolate very well, even though the learned amplitude is a little different from the ground truth, it has grasped the correct frequency of the underlying periodic function, both for the interpolation regime and the extrapolation regime. This shows that the proposed method has the desired flexibility towards periodicity, and has the potential to model such problems.

# 4 "Universal Extrapolation Theorem"

In contrast to the well-known universal approximation theorems [19, 8, 12] that qualifies a neural network on a bounded region, we prove a theorem that we refer to as the universal extrapolation theorem, which focuses on the behavior of a neural network with Snake on an unbounded region. This theorem says that a Snake neural network with a sufficient width can approximate any well-behaving periodic function.

**Theorem 3.** *Let $f(x)$ be a piecewise $C^1$ periodic function with period L. Then, a Snake neural network, $f_{w_N}$, with one hidden layer and with width N can converge to $f(x)$ uniformly as $N \to \infty$, i.e., there exists parameters $w_N$ for all $N \in \mathbb{Z}^+$ such that*

$$f(x) = \lim_{N \to \infty} f_{w_N}(x) \tag{5}$$

*for all $x \in \mathbb{R}$, i.e., the convergence is point-wise. If $f(x)$ is continuous, then the convergence is uniform.*

As a corollary, this theorem implies the classical approximation theorem [19, 32, 9], which states that a neural network with certain non-linearity can approximate any continuous function on a bounded region.

**Corollary 1.** *Let $f(x)$ be a two-layer neural network parametrized by two weight matrices $W_1$ and $W_2$, and let $w$ be the width of the network, then for any bounded and continuous function $g(x)$ on $[a, b]$, there exists $m$ such that for any $w \geq m$, we can find $W_1$, $W_2$ such that $f(x)$ is $\epsilon$-close to $g(x)$.*

This shows that the proposed activation function is a more general method than the ones previously studied because it has both (1) approximation ability on a bounded region and (2) the ability to learn periodicity on an unbounded region. The practical usefulness of our method is demonstrated experimentally to be on par with standard tasks, and to outperform previous methods significantly on learning periodic functions. Notice that the above theorem not only applies to Snake but also to the basic periodic functions such as $\sin$ and $\cos$ and monotonic variants such as $x + \sin(x)$, $x + \cos(x)$ etc.. While we argued for a preference towards $x + \sin^2(x)$, it remains to be determined by large-scale experiments in the industry to decide which one amongst these variants work better in practice, and we do encourage the practitioners to experiment with a few variants to decide the best suitable form for their application.

# 5 Initialization for Snake

As shown in [16], different activation functions actually require different initialization schemes (in terms of the sampling variance) to make the output of each layer unit variance, thus avoiding divergence or vanishing of the forward signal. Let $W \in \mathbb{R}^{d_1 \times d_2}$, whose input activations are $h \in \mathbb{R}^{d_2}$ with a unit variance for each of its element, and the goal is to set the variance of each element in $W$ such that $\text{Snake}(Wx)$ has a unit variance. To leading order, Snake looks like an identity function, and so one can make this approximation in finding the required variance: $\mathbb{E}\left(\sum_j^{d_2} W_{ij} x_j\right)^2 = 1$ which gives $\mathbb{E}[W_{ij}^2] = 1/\sqrt{d}$. If we use uniform distribution to initialize $W$, then we should sample from $Uniform(-\sqrt{\frac{3}{d}}, \sqrt{\frac{3}{d}})$, which is a factor of $\sqrt{2}$ smaller in range than the Kaiming uniform initialization. We notice that this initialization is often sufficient. However, when higher order correction is necessary, we provide the following exact solution, which is a function of $a$ in general.

**Proposition 1.** *The variance of expected value of $x + \frac{\sin^2(ax)}{a}$ under a standard normal distribution is $\sigma_a^2 = 1 + \frac{1 + e^{-8a^2} - 2e^{-4a^2}}{8a^2}$, which is maximized at $a_{max} \approx 0.56045$.*

The second term can be thought of as the "response" to the non-linear term $\sin^2(x)$. Therefore, one should also correct an additional bias induced by the $\sin^2(x)$ term by dividing the post-activation value by $\sigma_a$. Since the positive effect of this correction is the most pronounced when the network is deep, we compare the difference between having such a correction and having no correction on ResNet-101 on CIFAR-10. The results are presented in the appendix Section A.6.1. We note that using the correction leads to better training speed and better converged accuracy. We find that for standard tasks such as image classification, setting $0.2 \leq a \leq a_{max}$ to work very well. We thus set the default value of $a$ to be $0.5$. However, for tasks with expected periodicity, larger $a$, usually from 5 to 50 tend to work well.

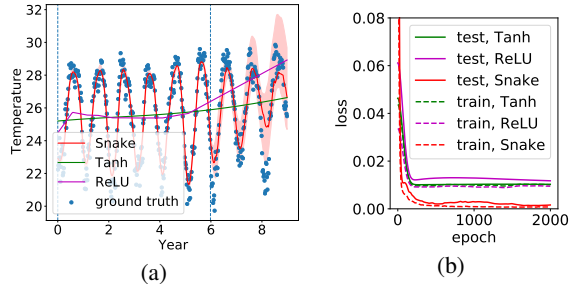

(a)                                    (b)

Figure 6: Experiment on the atmospheric data. (a) Regressing the mean weekly temperature evolution of Minamitorishima with different activation functions. For Snake, $a$ is treated as a learnable parameter and the red contour shows the 90% credibility interval. (b) Comparison of $\tanh$, ReLU, and Snake on a regression task with learnable $a$.

# 6 Applications

In this section, we demonstrate the wide applicability of Snake. We start with a standard image classification task, where Snake is shown to perform competitively against the popular activation functions, showing that Snake can be used as a general activation function. We then focus on the tasks where we expect Snake to be very useful, including temperature and financial data prediction. Training in all experiments are stopped at the time when the training loss of all the methods stop to decrease and becomes a constant. The performances of the converged model is not visibly different from the early stopping point for the tasks we considered, and the result would have been similar if we had chosen the early stopping point for comparison[1].

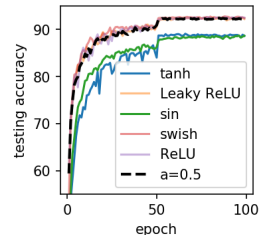

Figure 5: Comparison with other activation functions on CIFAR10.

## 6.1 Image Classification

*Experiment Description.* We train ResNet-18 [17], with roughly 10M parameters, on the standard CIFAR-10 dataset. We simply replace the activation functions in ReLU with the specified ones for comparison. CIFAR-10 is a 10-class image classification task of $32 \times 32$ pixel images; it is a standard dataset for measuring progress in modern computer vision methods[2]. We use LaProp [46] with the given default hyperparameters as the optimizer. We set learning rates to be $4e-4$ for the first $50$ epochs, and $4e-5$ for the last $50$ epochs. The standard data augmentation technique such as random crop and flip are applied. We note that our implementation reproduces the standard performance of ResNet18 on CIFAR-10, around $92-93\%$ testing accuracy. This experiment is designed to test whether Snake is suitable for standard and large-scale tasks one encounters in machine learning. We also compare this result against other standard or recently proposed activation functions including $\tanh$, ReLU, Leaky$-$ReLU [42], $Swish$ [34], and $\sin$ [30].

*Result and Discussion.* See Figure 5. We see that $\sin$ shows similar performance to $\tanh$, agreeing with what was found in [30], while Snake shows comparative performance to ReLU and Leaky$-$ReLU both in learning speed and final performance. This hints at the generality of the proposed method, and may be used as a replacement for ReLU in a straightforward way. We also test against other baselines on ResNet-101, which has 4 times more parameters than ResNet-18, to check if Snake can scale up to even larger and deeper networks, and we find that, consistently, Snake achieves similar performance ($94.1\%$ accuracy) to the most competitive baselines.

## 6.2 Atmospheric and Body Temperature Prediction

For illustration, we first show two real-life applications of our method to predicting the atmospheric temperature of a local island, and human body temperature. These can be very important for medical applications. Many diseases and epidemics are known to have strong correlation with atmospheric temperature, such as SARS [6] and the current COVID-19 crisis ongoing in the world [44, 36, 27]. Therefore, being able to model temperature accurately could be important for related policy making.

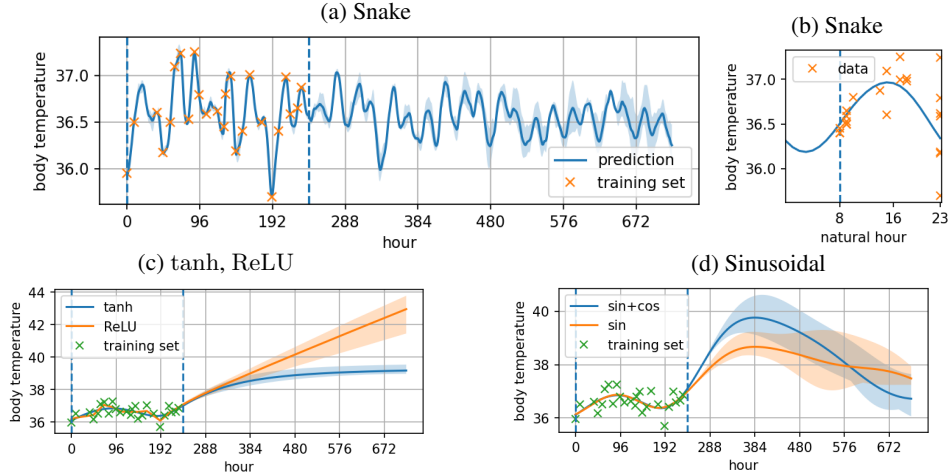

Figure 7: Prediction of human body temperature. (a) Snake; (b) Averaged temperature prediction in a circadian cycle (i.e. as a function of the natural hours in a day); (c) $\tanh$ and ReLU; (d) sinusoidal activations.

**Atmospheric temperature prediction.** We start with testing a feedforward neural network with two hidden layers (both with 100 neurons) to regress the temperature evolution in Minamitorishima, an island south of Tokyo (longitude: 153.98, latitude: 24.28)[3]. The data represents the average weekly temperature after April 2008 and the results are shown in Fig. 6a. We see the $\tanh$ and ReLU-based models fail to optimize this task, and do not make meaningful extrapolation. On the other hand, the Snake-based model succeeds in optimizing the task and makes meaningful extrapolation with correct period. Also see Figure 6b. We see that Snake achieves vanishing training loss and generalization loss, while the baseline methods all fail to optimize to 0 training loss, and the generalization loss is also not satisfactory. We also compare with the more recent activation functions such as the Leaky-ReLU and Swish, and similar results are observed; see appendix.

**Human body temperature.** Modeling the human body temperature may also be important; for example, fever is known as one of the most important symptom signifying a contagious condition, including COVID19 [15, 39]. *Experiment Description.* We use a feedforward neural network with 2 hidden layers, (both with 64 neurons) to regress the human body temperature. The data is measured irregularly from an anonymous participant over a 10-days period in April, 2020, of 25 measurements in total. While this experiment is also rudimentary in nature, it reflects a great deal of obstacles the community faces, such as very limited (only 25 points for training) and insufficient measurement taken over irregular intervals, when applying deep learning to real problems such as medical or physiological prediction [18, 33]. In particular, we have a dataset where data points from certain period in a day is missing, for example, from 12am to 8am, when the participant is physically at rest (See Figure 7b), and for those data points we have, the intervals between two contiguous measurements are irregular with 8 hours being the average interval, yet this is often the case for medical data where exact control over variables is hard to realize. The goal of this task is to predict the body temperature at *at every hour*. The model is trained with SGD with learning rate $1e-2$ for 1000 steps, $1e-3$ for another 1000 steps, and $5e-4$ for another 1000 steps.

*Results and Discussion.* The performances of using ReLU, $\tanh$, sin, sin + cos and Snake are shown in Figure 7. We do not have a testing set for this task, since it is quite unlikely that a model will predict correctly for this problem due to large fluctuations in human body temperature, and we compare the results qualitatively. In fact, we have some basic knowledge about body temperature. For example, (1) it should fall within a reasonable range from 35.5 to 37.5 Celsius degree [13], and, in fact, this is the range where all of the training points lie; (2) at a finer scale, the body temperature follows a periodic behavior, with highest in the afternoon (with a peak at around 4pm), and lowest in the midnight (around 4am) [13]. At the bare minimum, a model needs to obey (1), and a reasonably well-trained model should also discover (2). However, $\tanh$ or ReLU fail to limit the temperature to the range 35.5 and 37.5 degree. Both baselines extrapolate to above 39 degree at 20 days beyond the training set. In contrast, learning with Snake as the activation function learned to obey the first rule.

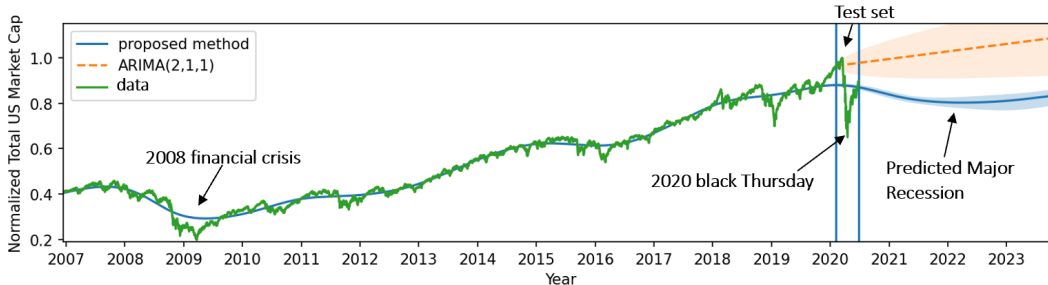

Figure 8: Prediction of Wilshire 5000 index, an indicator of the US and global economy.

See Figure 7.a. To test whether the model has also grasped the periodic behavior specified in (2), we plot the average hourly temperature predicted by the model over a 30 days period. See Figure 7b. We see that the model does capture the periodic oscillation as desired, with peak around 16pm and minimum around 4am. The successful identification of 4am is extremely important, because this is in the range where no data point is present, yet the model inferred correctly the periodic behavior of the problem, showing Snake really captures the correct inductive bias for this problem.

## 6.3 Financial Data Prediction

*Problem Setting.* The global economy is another area where quasi-periodic behaviors might happen [21]. At microscopic level, the economy oscillates in a complex, unpredictable manner; at macroscopic level, the global economy follows a $8 - 10$ year cycle that transitions between periods of growth and recession [5, 37]. In this section, we compare different models to predict the total US market capitalization, as measured by the Wilshire 5000 Total Market Full Cap Index[4] (we also did the same experiment on the well-known Buffet indicator, which is seen as strong indicator for predicting national economic trend [24]; we also see similar results). For training, we take the daily data from 1995-1-1 to 2020-1-31, around 6300 points in total, the ending time is deliberately chosen such that it is before the COVID19 starts to affect the global economy [3, 11]. We use the data from $2020 - 2 - 1$ to $2020 - 5 - 31$ as the test set. Noticeably, the test set differs from training set in two ways (1) a market crush called black Thursday happens (see Figure 8); (2) the general trend is recessive (market cap moving downward on average). It is interesting to see whether the bearish trend in this period is predictable without the affect of COVID19. For neural network based methods, we use a $4$-layer feedforward network with $1 \to 64 \to 64 \to 1$ hidden neurons, with specified activation function, we note that no activation function except Snake could optimize to vanishing training loss. The error is calculated with $5$ runs.

*Results and Discussion.* See Table 2, we see that the proposed method outperforms the competitors by a large margin in predicting the market value from 2020-2-1. Qualitatively, we focus on making comparison with ARIMA, a traditional and standard method in economics and stock price prediction [29, 2, 41]. See Figure 8. We note that ARIMA predicts a growing economy, Snake predicts a recessive economy from 2020-2-1 onward. In fact, for all the methods in Table 2, the proposed method is the only method that predicts a recession in and beyond the testing period, we hypothesize that this is because the proposed method is only method that learns to capture the long term economic cycles in the trend. Also, it is interesting that

| Method | MSE on Test Set |
|---|---|
| ARIMA$(2, 1, 1)$ | $0.0215^{\pm 0.0075}$ |
| ARIMA$(2, 2, 1)$ | $0.0306^{\pm 0.0185}$ |
| ARIMA$(3, 1, 1)$ | $0.0282^{\pm 0.0167}$ |
| ARIMA$(2, 1, 2)$ | $0.0267^{\pm 0.0154}$ |
| ReLU DNN | $0.0113^{\pm 0.0002}$ |
| Swish DNN | $0.0161^{\pm 0.0007}$ |
| $\sin + \cos$ DNN | $0.0661^{\pm 0.0936}$ |
| $\sin$ DNN | $0.0236^{\pm 0.0020}$ |
| Snake | $\mathbf{0.0089^{\pm 0.0002}}$ |

Table 2: Prediction of Wilshire 5000 Index from 2020-2-1 to 2020-5-31.

the model predicts a recession without predicting the violent market crash. This might suggest that the market crash is due to the influence of COVID19, while a simultaneous background recession also occurs, potentially due to global business cycle. For purely analysis purpose, we also forecast the prediction until 2023 in Figure 8. Alarmingly, our method predicts a long-term global recession, starting from this May, for an on-average 1.5 year period, only ending around early 2022. This also suggests that COVID19 might not be the only or major cause of the current recessive economy.

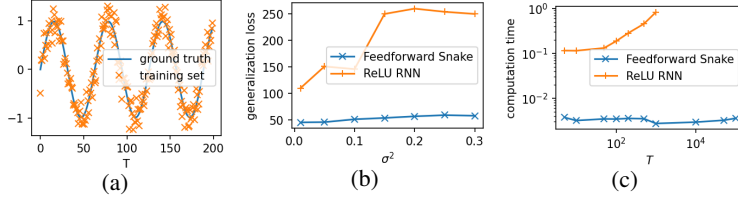

Figure 9: Comparison between simple RNN (with ReLU as activation function) and feedforward network with Snake as activation function. (a) The task is to regress a length-$T$ time series generated by a simple periodic function contaminated with a white gaussian noise with variance $\sigma^2$. (b) As the noise increases, the RNN fails in generalization, while this has relatively no effect on the proposed method. (c) One more important advantage of the proposed method is that it requires much less computation time during forward and backward propagation.

### 6.4 Comparison with RNN on Regressing a Simple Periodic Function

One important application of the learning of periodicities is time series prediction, where the periodicity is at most one-dimensional. This kind of data naturally appears in many audio and textual tasks, such as machine translation [4], audio generation [14], or multimodal learning [23]. Therefore, it is useful to compare with RNN on related tasks. However, we note that the problem with RNNs is that they implicitly parametrize the data point $x$ by time: $x = x(t)$. It is hence limited to model periodic functions of at most 1d and cannot generalize to a periodic function of arbitrary dimension, which is not a problem for our proposed method. We perform a comparison of RNN with Snake deployed on a feedforward network on a 1d problem. See Figure 9.a for the training set of this task. The simple function we try to model is $y = \sin(0.1x)$, we add a white noise with variance $\sigma^2$ to each $y$, and the model sees a time series of length $T$. See 9.b for the performance of both models, when $T = 100$, and validated on a noise-free hold-out section from $t = 101$ to $300$. We see that the proposed method outperforms RNN significantly. On this task, One major advantage of our method is that it does not need to back-propagate through time (BPTT), which both causes vanishing gradient and prohibitively high computation time during training [31]. In Figure 9.c we plot the average computation time of a single gradient update vs. the length of the time series, we see that, even at smallest $T = 5$, the RNN requires more than 10 times of computation time to update (when both models have a similar number of parameters, about 3000). For Snake, the training is done with gradient descent on the full batch, and the computation time remains low and does not increase visibly as long as the GPU memory is not overloaded. This is a significant advantage of our method over RNN. Snake can also be used in a recurrent neural network, and is also observed to improve upon $\mathrm{ReLU}$ and $\tanh$ for predicting long term periodic time evolution. Due to space constraint, we discuss this in section A.2.

## 7 Conclusion

In this work, we have identified the extrapolation properties as a key ingredient for understanding the optimization and generalization of neural networks. Our study of the extrapolation properties of neural networks with standard activation functions suggest the lack of capability to learn a periodic function: due to the mismatched inductive bias, the optimization is hard and generalization beyond the range of observed data points fails. We think that this example suggests that the extrapolation properties of a learned neural networks should deserve much more attention than it currently receives. We then propose a new activation function to solve this periodicity problem, and its effectiveness is demonstrated through the "extrapolation theorem", and then tested on standard and real-life application experiments. We also hope that our current study will attract more attention to the study of modeling periodic functions using deep learning.

## Acknowledgement

This work was supported by KAKENHI Grant No. JP18H01145, 19K23437, 20K14464, and a Grant-in-Aid for Scientific Research on Innovative Areas "Topological Materials Science (KAKENHI Grant No. JP15H05855) from the Japan Society for the Promotion of Science. Liu Ziyin thanks the graduate school of science of the University of Tokyo for the financial support he receives. He also thanks Zhang Jie from the depth of his heart; without Zhang Jie's help, this work could not have

been finished so promptly. We also thank Zhikang Wang and Zongping Gong for offering useful discussions.

## Broader Impact Statement

In the field of deep learning, we hope that this work will attract more attention to the study of how neural networks extrapolate, since how a neural network extrapolates beyond the region it observes data determines how a network generalizes. In terms of applications, this work may have broad practical importance because many processes in nature and in society are periodic in nature. Being able to model periodic functions can have important impact to many fields, including but not limited to physics, economics, biology, and medicine.

## Footnotes

[1]The code for our implementation of a demonstrative experiment can be found at `https://github.com/AdenosHermes/NeurIPS_2020_Snake`.

[2]Our code is adapted from `https://github.com/kuangliu/pytorch-cifar`

[3]Data from `https://join.fz-juelich.de/access`

[4]Data from https://www.wilshire.com/indexes

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
