[Supplementary Material]

# A  Additional Experiments

## A.1  Effect of using different $a$ and More Periodic Function Regressions

It is interesting to study the behavior of the proposed method on different kinds of periodic functions (continuous, discontinuous, compound periodicity). See Figure 10. We see that using different $a$ seems to bias model towards different frequencies. Larger $a$ encourages learning with larger frequency and vice versa. For more complicated periodic functions, see Figure 11 and 12.

(a) Snake, $a = 5.5$

(b) Snake, $a = 10$

(c) Snake, $a = 30$

Figure 10: Effect of using different $a$.

Figure 11: Regressing a rectangular function with Snake as activation function for different values of $a$. For a larger value of $a$, the extrapolation improves.

Figure 12: Regressing $\sin(x) + \sin(4x)/4$ with Snake as activation function for different values of $a$. For a larger value of $a$, the extrapolation improves: Whereas the $a = 1$-model treats the high-frequency modulation as noise, the $a = 16$-model seems to learn a second signal with higher frequency (bottom centre).

Figure 13: Regressing a simple sin function with tanh, ReLU, and Snake as the activation function.

## A.2    Learning a Periodic Time Evolution

In this section, we try to fit a periodic dynamical system whose evolution is given by $x(t) = \cos(t/2) + 2\sin(t/3)$, and we use a simple recurrent neural network as the model, with the standard $\mathrm{tanh}$ activation replaced by the designated activation function. We use Adam as the optimizer. See Figure 13. The region within the dashed vertical lines are the range of the training set.

(a) Prediction. Learning range is indicated by the blue vertical bars.

(b) Learning loss during training.

Figure 14: Comparison between Snake, tanh, and ReLU as activation functions to regress and predict the EUR-USD exchange rate.

## A.3 Currency exchange rate modelling

We investigate how Snake performs on one more financial data. The task is to predict the exchange rate between EUR and USD. As in the main text, we use a two-hidden-layer feedforward network with 256 neurons in the first and 64 neurons in the second layer. We train with SGD, with a learning rate of $10^{-4}$, weight decay of $10^{-4}$, momentum of $0.99$, and a mini-batch size of $16^4$. For Snake, we make $a$ a learnable parameter. The result can be seen in Fig. 14. Only Snake can model the rate on the training range and makes the most realistic prediction for the exchange rate beyond the year 2015. The better optimization and generalization property of Snake suggests that it offers the correct inductive bias to model this task.

Figure 15: Full Training set for Section 6.3

Figure 16: Learning trajectory of Snake. One notices that Snake firsts learns linear features, then low frequency features and then high frequency features.

## A.4 How does Snake learn?

We take this chance to study the training trajectory of Snake using the market index prediction task as an example. We set $a = 20$ in this task. See Figure 15 for the full training set for this section (and also for Section 6.3). See Figure 16 for how the learning proceeds. Interestingly, the model first learns an approximately linear function (at epoch 10), and then it learns low frequency features, and then learns the high frequency features. In many problems such as image and signal processing [1], the high frequency features are often associated with noise and are not indicative of the task at hand. This experiment explains in part the good generalization ability that Snake seems to offer. Also, this suggests that one can also devise techniques to early stopping on Snake in order to prevent the learning of high-frequency features when they are considered undesirable to learn.

(a) training loss

(b) testing accuracy

Figure 17: Comparison on CIFAR10 with ResNet18. We see that for a range of choice of $a$, there is no discernible difference between the generalization accuracy of ReLU and Snake.

Figure 18: Grid Search for Snake at different $a$ on ResNet18.

Figure 19: ResNet100 on CIFAR-10. We see that the proposed method achieves comparable performance to the ReLU-style activation functions, significantly better than tanh and sin.

## A.5 CIFAR-10 Figures

In this section, we show that the proposed activation function can achieve performance similar to ReLU, the standard activation function, both in terms of generalization performance and optimization speed. See Figure 17. Both activation functions achieve $93.5 \pm 1.0\%$ accuracy.

## A.6 CIFAR-10 with ResNet101

To show that Snake can scale up to larger and deep neural networks, we also repeat the experiment on CIFAR-10 with ResNet101. See Figure 19. Again, we see that the Snake achieves similar performance to ReLU and Leaky-ReLU.

(a) training loss vs. epoch        (b) testing accuracy vs. epoch

Figure 20: Effect of variance correction

### A.6.1 Effect of Variance Correction

In this section, we show the effect of variance correction is beneficial. Since the positive effect of correction is the most pronounced when the network is deep, we compare the difference between having such correction and having no correction on ResNet101 on CIFAR-10. See Figure 20; we note that using the correction leads to better training speed and better converged accuracy.

We also restate the proposition here.

**Proposition 2.** *The variance of expected value of $x + \frac{\sin^2(ax)}{a}$ under a standard normal distribution is $\sigma_a^2 = 1 + \frac{1 + e^{-8a^2} - 2e^{-4a^2}}{8a^2}$, which is maximized at $a_{max} \approx 0.56045$.*

*Proof.* The proof is straight-forward calculation. The second moment of Snake is $1 + \frac{3 + e^{-8a^2} - 4e^{-2a^2}}{8a^2}$, while the squared first moment is $\frac{e^{-4a^2}(-1 + e^{2a^2})^2}{4a^2}$, and subtracting the two, we obtain the desired variance

$$\sigma_a^2 = 1 + \frac{1 + e^{-8a^2} - 2e^{-4a^2}}{8a^2}. \tag{6}$$

Solving for this numerically from a numerical solver (we used Mathematica) renders the maximum at $a_{max} \approx 0.56045$. $\square$

# B Proofs for Section 2.2

We reproduce the statements of the theorems for the ease of reference.

**Theorem 4.** *Consider a feed forward network $f_{\text{ReLU}}(x)$, with arbitrary but fixed depth $h$ and widths $d_1, ..., d_{h+1}$. Then*

$$\lim_{z \to \infty} \|f_{\text{ReLU}}(zu) - zW_u u - b_u\|_2 = 0, \tag{7}$$

*where $z$ is a real scalar, $u$ is any unit vector of dimension $d_1$, and $W_u \in \mathbb{R}^{d_1 \times d_h}$ is a constant matrix only dependent on $u$.*

We prove this by induction on $h$. We first prove the base case when $h = 2$, i.e., a simple non-linear neural network with one hidden layer.

**Lemma 1.** *Consider feed forward network $f_{\text{ReLU}}(x)$ with $h = 2$. Then*

$$\lim_{z \to \infty} \|f_{\text{ReLU}}(zu) - zW_u u - b_u\|_2 = 0 \tag{8}$$

*for all unit vector $u$.*

*Proof.* In this case,

$$f_{\text{ReLU}}(x) = W_2 \sigma(W_1 x + b_1) + b_2,$$

where $\sigma(x) = \text{ReLU}(x)$, and let $\mathbf{1}_{x>0}$ denote the vector that is 1 when $x > 0$ and zero otherwise, and let $M_{x>0} := \text{diag}(\mathbf{1}_{x>0})$, then for any fixed $u$ we have

$$f_{\text{ReLU}}(zu) = W_2 M_{W_1 zu + b_1 > 0}(W_1 zu + b_1) + b_2 = zW_u u + b_u \tag{9}$$

where $W_u := W_2 M_{W_1 zu + b_1 > 0} W_1$ and $b_u = W_u b_1 + b_2$, and $W_u$ is the desired linear transformation and $b_u$ the desired bias; we are done. □

Apparently, due the self-similar structure of a deep feedforward network, the above argument can be iterated over for every layer, and this motivates for a proof based on induction.

*Proof of Theorem.* Now we induce on $h$. Let the theorem hold for any $h \le n$, and we want to show that it also holds for $h = n + 1$. Let $h = n + 1$, we note that any $f_{\text{ReLU},\ h=n+1}$ can be written as

$$f_{\text{ReLU},\ h=n+1}(zu) = f_{\text{ReLU},\ h=2}(f_{\text{ReLU},\ h=n}(zu)) \tag{10}$$

then, by assumption, $f_{\text{ReLU},\ h=n}(x)$ approaches $zW_u u + b_u$ for some linear transformation $W_u$, $b_u$:

$$\lim_{z \to \infty} f_{\text{ReLU},\ h=n+1}(zu) = f_{\text{ReLU},\ h=2}(zW_u u + b_u) \tag{11}$$

and, by the lemma, this again converge to a linear transformation, and we are done. □

Now we can prove the following theorem, this proof is much simpler and does not require induction.

**Theorem 5.** *Consider a feed forward network $f_{\tanh}(x)$, with arbitrarily fixed depth $h$ and widths $d_1, ..., d_{h+1}$, then*

$$\lim_{z \to \infty} \|f_{\text{ReLU}}(zu) - v_u\|_2 = 0, \tag{12}$$

*where $z$ is a real scalar, and $u$ is any unit vector of dimension $d_1$, and $v_u \in \mathbb{R}^{d_{h+1}}$ is a constant vector only depending on $u$.*

*Proof.* It suffices to consider a two-layer network. Likewise, $f_{\tanh}(zu) = (W_2 \sigma(W_1 zu + b_1) + b_2)$, where $\sigma(x) = \tanh(x)$. As $z \to \infty$, $W_1 zu + b_1$ approaches either positive or negative infinity, and so $\sigma(W_z zu + b_1)$ approaches a constant vector whose elements are either 1 or $-1$, which is a constant vector, and $(W_z \sigma(W_z zu + b_1) + b_2)$ also approaches some constant vector $v_u$.

Now any layer that are composed after the first hidden layer takes in an asymptotically constant vector $v_u$ as input, and since the activation function $\tanh$ is a continuous function, $f_{\tanh}(x)$ is continuous, and so

$$\lim_{z \to \infty} f_{\tanh,\ h=n}(x) = f_{\tanh,\ h=n-1}(v_u) = v_u'. \tag{13}$$

We are done. □

## C   Universal Extrapolation Theorems

**Theorem 6.** *Let $f(x)$ be a piecewise $C^1$ periodic function with period L. Then, a Snake neural network, $f_{w_N}$, with one hidden layer and with width $N$ can converge to $f(x)$ uniformly as $N \to \infty$, i.e., there exists parameters $w_N$ for all $N \in \mathbb{Z}^+$ such that*

$$f(x) = \lim_{N \to \infty} f_{w_N}(x) \tag{14}$$

*for all $x \in \mathbb{R}$, i.e., the convergence is point-wise. If $f(x)$ is continuous, then the convergence is uniform.*

We first show that using $\cos$ as activation function may approximate any periodic function, and then show that Snake may represent a $\cos$ function exactly.

**Lemma 2.** *Let $f(x)$ be defined as in the above theorem and let $f_{w_N}(x)$ be a feedforward neural network with $\cos$ as the activation function, then $f_{w_N}(x)$ can converge to $f(x)$ point-wise.*

*Proof.* It suffices to show that a neural network with $\sin$ as activation function can represent a Fourier series to arbitrary order, and then apply the Fourier convergence theorem. By the Fourier convergence theorem, we know that

$$f(x) = \frac{a_0}{2} + \sum_{m=1}^{\infty} \left[ \alpha_m \cos\left(\frac{m\pi x}{L}\right) + \beta_m \sin\left(\frac{m\pi x}{L}\right) \right], \tag{15}$$

for unique Fourier coefficients $\alpha_m$, $\beta_m$, and we recall that our network is defined as

$$f_{w_N}(x) = \sum_{i=1}^{D} w_{2i} \cos(w_{1i}x + b_{1i}) + b_{2i} \tag{16}$$

then we can represent Eq. 15 order by order. For the $m$-th order term in the Fourier series, we let

$$w_{2,2m-1} = \beta_m = \int_{-L}^{L} f(x) \sin\left(\frac{m\pi x}{L}\right) dx \tag{17}$$

$$w_{1,2m-1} = \frac{m\pi}{L} \tag{18}$$

$$w_{2,2m} = \alpha_m = \int_{-L}^{L} f(x) \cos\left(\frac{m\pi x}{L}\right) dx \tag{19}$$

$$w_{1,2m} = \frac{m\pi}{L} \tag{20}$$

$$b_{1,2m-1} = -\frac{\pi}{2} \tag{21}$$

$$\tag{22}$$

and let the unspecified biases $b_i$ be 0: we have achieved an exact parametrization of the Fourier series of $m$ order with a $\sin$ neural network with $2m$ many hidden neurons, and we are done. $\square$

The above proof is done for a $\cos(x)$ activation; we are still obliged to show that Snake can approximate a $\cos(x)$ neuron.

**Lemma 3.** *A finite number of Snake neurons can represent a single $\cos$ activation neuron.*

*Proof.* Since the frequency factor $a$ in Snake can be removed by a rescaling of the weight matrices, we set $a = 2$, and so $x + \sin^2(x) = x - \cos(x) + \frac{1}{2}$. We also reverse the sign in front of $\cos$, and remove the bias $\frac{1}{2}$, and prove this lemma for $x + \cos(x)$. We want to show that for a finite $D$, there $w_1$ and $w_2$ such that

$$\cos(x) = \sum_{i=1}^{D} w_{2,i}(w_{1,i}x + b_{1,i}) + b_{2,i} + \sum_{i=1}^{D} w_{2,i} \cos(w_{1,i}x + b_{1,i}) \tag{23}$$

This is achievable for $D = 2$, let $w_{1,1} = -w_{1,2} = 1$, and let $b_{1,i} = b_{2,i} = 0$, we have:

$$\cos(x) = (w_{2,1} - w_{2,2})x + \sum_{i=1}^{D} w_{2,i} \cos(x) \tag{24}$$

and set $w_{2,1} = w_{2,2} = \frac{1}{2}$ achieves the desired result. Combining with the result above, this shows that a Snake neural network with $4m$ many hidden neurons can represent exactly a Fourier series to $m$-th order. □

**Corollary 2.** *Let $f(x)$ be a two layer neural network parametrized by two weight matrices $W_1$ and $W_2$, and let $w$ be the width of the network, then for any bounded and continuous function $g(x)$ on $[a,b]$, there exists $m$ such that for any $w \geq m$, we can find $W_1, W_2$ such that $f(x)$ is $\epsilon$−close to $g(x)$.*

*Proof.* This follows immediately by setting $[a,\ b]$ to match the $[-L,\ L]$ region in the previous theorem. □

## Footnotes

[4]Hyperparameter exploration performed with Hyperactive: `https://github.com/SimonBlanke/Hyperactive`