[Reviews · NeurIPS 2020]

Review 1

Summary and Contributions: # UPDATE I have read all the rebuttal and other reviews and after the discussion I would like to still recommend acceptance of this paper (7). # REVIEW This paper studies the problem of extrapolation in Neural Networks. In general, extrapolation problem is ill defined, as we ask a model to predict results outside its training regime. However, there are inductive biases that govern many aspects of real life, such as periodicity, that authors focus on in this paper. The main contribution is to propose a simple activation function, that on one hand provided periodicity bias, and on the other avoids issues of training with it that has been encountered in numerous works in the past using sin or cosine activations.

Strengths: - authors identify and propose unified solution for introducing periodicity bias and avoiding typical optimisation issues of this challenge - proposed solution is extremely simple and ready to use out of the box - authors provide simple theoretical foundations of their approach, in an easy to digest way - proposed method is evaluated on a wide range of tasks from different disciplines, including those not having a periodic structure

Weaknesses: - empirical validation could use more examples of clearly non-periodic problems apart from basic computer vision task, but this should not be seen as an argument against accepting this paper, but rather an a valuable addon that would provide even better picture of the proposed solution and test it wide applicability - presentation of proofs in the appendix is of very low quality. It is full of typos, non-gramatical statements, logical issues (authors talk about representing sine and then write a lemma about cosine). These do not lead to invalid results, but look extremely rushed and of low quality.

Correctness: - please consider renaming "Universal Extrapolation Theorem" to "Universal Periodic Extrapolation Theorem" for correctness

Clarity: Overall paper is well written, with small presentation issues (e.g. why is Function capitalised? There are some extra white spaces floating around)

Relation to Prior Work: Paper cites related prior work.

Reproducibility: Yes

Additional Feedback:


Review 2

Summary and Contributions: After rebuttal, I somewhat disagree with the author's disagreement with regards to RNNs. Although I agree that 2D or higher periodic functions are certainly not handled with RNNs, I point out the ubiquity of 1D (i.e. time) periodic functions. So much so that indeed the authors have validated against solely 1D periodic functions. I am hoping the authors will consider 1D periodic as a special case and address comparisons against RNNs more thoroughly. For example, another reviewer suggested the computational efficiency and vanishing gradient problem likely to occur for periodic functions with a large wavelength. The paper proposes a novel activation function to model data with [possible] periodic elements. The idea is justified with some theoretical proofs demonstrating the inability of traditional activation functions to capture periodic data. Validation is performed on several datasets containing periodic elements showing empirically that indeed the activation function is able to learn periodic functions better than traditional activation functions.

Strengths: This paper identifies a significant shortcoming in traditional feedforward networks, in particular their inability to generalize beyond a compact domain. That being said, this is by design as the data is assumed to be i.i.d. with the training dataset sampled from the same distribution as the test dataset. Alternatively this paper considers i.i.d. sampled data in fourier space (restricted to a compact region in R^d), as opposed to R^d space. The intention of this is to allow generalization beyond the compact set within which the training data lies. A straightforward solution is provided to capture this shortcoming with the aid of a particular activation function of a periodic nature. The universal approximation theorem is extended to this activation function. A decent amount of validation is performed with respect to this activation function.

Weaknesses: A significant shortcoming in the approach is the lack of proper and thorough validation with recurrent neural networks. The stated problem (i.i.d. in fourier space, restricted to a compact region in R^d) can be tackled with the auto-regressive approach provided by RNNs. However this is not mentioned, or compared against in the paper. I note that RNNs are somewhat compared against in the appendix, but this is with respect to RNN+Snake vs RNN. Above all this paper needs a comparison showing the ability of feedforward + Snake networks to outperform vanilla RNNs. Beyond this, RNNs should also make up at least some of the related work such that the two competing approaches (autoregressive vs feedforward + Snake) are well compared and contrasted. Although I imagine that this approach *does* solve some of the shortcomings of RNNs (i.e. vanishing gradient across long sequences), a thorough validation and comparison is needed to verify this. Section 3 text should be significantly improved. I found the rather vague and imprecise argument with respect to alternatives unconvincing. It's difficult to evaluate the merits of a mathematical argument which is not formally stated. I also remain unconvinced by the presence of second order terms in the taylor expansion around 0 to be sufficient to explain improvement. This should be formally or empirically explained in more detail. (e.g. by validating against the specific taylor expansion with/without the second order term).

Correctness: I agree with the authors in the shortcoming of traditional feedforward networks to capture periodic data. I somewhat question the inclusion of the 'universality' guarantees provided by this new activation function. I think Cybenko's result can be easily extended to almost any activation function. Perhaps this belongs solely in the appendix with further validation in the main text. I found the financial data prediction validation somewhat less than convincing. With regards to the current health situation, it is not yet clear whether current conditions are 'too remarkable' to be admitted as a suitable test set. Perhaps validation with previous recessions would have been a better idea.

Clarity: The paper is well written enough for my taste.

Relation to Prior Work: This paper is framed 'ok', with comparison provided to the most directly related work. However, I would have appreciated a more broader comparison with respect to both Fourier networks, and RNNs.

Reproducibility: Yes

Additional Feedback: I think this paper is promising given additional work. In particular it must be significantly motivated and empirically validated why an auto-regressive approach (i.e. RNNs) do not work well in this case. However, without this specific validation, I cannot see this paper as ready for publication.


Review 3

Summary and Contributions: The researchers develop their Snake activation which they assert is a better activation function for learning periodic functions (such as sine waves).

Strengths: The problem of feed-forward network extrapolation on periodic functions is a well-known and important problem, which this research seeks to address. The authors provide good heuristics for initialization, which I believe is quite useful for researchers hoping to utilize this methodology on different classes of problems. The authors evaluate, I believe, on a varied set of problems, lending credence to the generalizability of this work.

Weaknesses: The methodology, although well explained, leaves too much uncertainty on the table as far as correctness. Specifically the lack of any discussion for the stopping criteria. It is not clear, at least from my read, whether the researchers cherry-picked their stopping criteria to give their algorithm the best results. I doubt this is what happened, sincerely. However, for NeurIPS quality papers, there should be no doubt about the correctness of the methodology for such basic tasks as stopping criteria.

Correctness: To expound a bit more on the weaknesses above-- For the first problem (3.1), using a set number of epochs to compare the different activation functions is a sub-optimal approach, as we do not know if any of the networks suffered from over-fitting. The authors should use validation loss and early stopping in this particular problem. I don't think the problem with extrapolation is due to over-fitting, however, as I have performed these experiments, myself, long ago, and this is a commonly known problem. Perhaps the authors believed this to be the case and took a shortcut here. For the other examples, it should be clear that the network is stopping based on validation loss, and not the authors just picking a stopping point that works for their model. They should also ensure that the parameters that led to that stopping point are used throughout and not cherry-picked per problem to yield the best results. Though I felt I was careful, it is possible that I somehow overlooked the authors use of correct validation set usage. If I have, please forgive me, and I would be amenable to changing my score.

Clarity: The paper is clearly written. There is a typo for the word "daily" on line 284.

Relation to Prior Work: The authors motivate their problem well. It would have been nice to see a comparison against "Phase-functioned neural networks for character control" https://doi.org/10.1145/3072959.3073663. Also, it would be nice to see RNNs in general compared to the efficacy of Snake in a standard FFN. Even if RNNs worked better, it still is, in my opinion, wonderful to see a unique solution to periodic problems using only FFNs.

Reproducibility: Yes

Additional Feedback: The list many parameters to their networks, including optimizer, epochs, etc. I think reproduction wouldn't be too difficult, sans the weaknesses due to validation set.


Review 4

Summary and Contributions: In this paper, a novel activation function - Snake – is proposed, aiming to account for extrapolation in periodic signals.

Strengths: (+) introducing a periodic and monotonic activation function (+) the research around new and better activation functions is very relevant

Weaknesses: (-) Some claims are way too strong. *“we study and prove the incapability of standard activation functions to extrapolate” is debatable. The proof is mainly related with ReLU and tanh. * After Corollary 1, row 164: “the proposed activation function is a more general method than the ones previously studied”. In what sense a periodic function is more general? (-) The main problem is the evaluation of the method. The paper does not show that the proposed method provides clear benefits against existing methods. Experiments do not show that Snake activation is better than the existing activation functions - ReLU and their variants (e.g. Leaky ReLU) on any benchmark database. From Figure 5, I can conclude that Swish and Leaky ReLU are comparable with the proposed method (or even better) for CIFAR-10. Thus, why these methods are not also used in the last two experiments (6.2 and 6.3) where Snake activation obtains the best results? The same omission is hold for MNIST (figure 3). The above reflects my understanding, and I may have missed something. But if I am correct, the experiments in this paper fail to demonstrate the usefulness of the method.

Correctness: The method is clear and the proof is mainly related whit ReLU and tanh. Overall, the claims are too strong and are not mirrored by the results. The empirical methodology is not consistent across the experiments.

Clarity: Overall a nice reading, with a good structure. The first part of the experiments (CIFAR 10) can be better embedded in the overall paper context. *Some notations are used in the paper and explained in the appendix. *Minor typos: fucntions -> functions; that the the wide -> that the wide

Relation to Prior Work: There is a clear distinction between the Snake function and other activation functions. Still, a large range of activations functions are even not mentioned, such as SReLU, GELU, ELU, SELU, RReLU. Furthermore, the authors should be aware of the previous work done on the periodic and monotonic activation functions. J. M. Sopena, E. Romero and R. Alquezar, "Neural networks with periodic and monotonic activation functions: a comparative study in classification problems," ICANN, 1999.

Reproducibility: Yes

Additional Feedback: In my opinion, the paper will strongly benefit if *the claims are revised *expand the previous literature (see above) *clarify the experiments, and ensure their consistence

[Author Response · NeurIPS 2020]

General Comment: We are very grateful to all the reviewers for carefully reading our paper. Their feedback has helped us to improve the paper accordingly. The two messages of this paper are that (1) learning a periodic or semi-periodic function with neural networks is yet unresolved, and we argue that the key to solve this is to focus on the extrapolation properties of neural networks; (2) we proposed to solve this problem with a simple alternative activation function.

**"A significant shortcoming is the lack of comparison with recurrent neural networks"** (R2, R4): We respectfully yet strongly disagree with this. It is indeed interesting to see how RNN would perform for these tasks. However, the problem with RNNs is that they implicitly parametrize the data point $x$ by time: $x = x(t)$. It is hence limited to model periodic functions of at most 1d and cannot generalize to a periodic function of arbitrary dimension; e.g., it is not clear how one could define RNN to learn the function $f(x, z) = \sin(x) + \sin(z)$, which is an easy task for feedforward networks with Snake. For this reason, we do not believe our method (with Snake + feedforward) needs to be a competitor to RNN. This being said, we perform a comparison of RNN with Snake with feedforward on a 1d problem. See Figure 1.a for the training set of this task. The simple function we try to model is $y = \sin(0.1x)$, we add a white noise with variance $\sigma^2$ to each $y$, and the model sees a time series of length $T$. See 1.b for the performance of both models, when $T = 100$, and validated on a noise-free hold-out section from $T = 101$ to $300$. We see that the proposed method outperforms RNN significantly. On this task, One major advantage of our method is that it does not need to back-propagate through time (BPTT), which both causes vanishing gradient and prohibitively high computation time during training. In Figure 1.c we plot the average computation time of a single gradient update vs. the length of the time series, we see that, even at smallest $T = 5$, the RNN requires more than 10 times of computation time to update (when both models have a similar number of parameters, about 3000). This is a significant advantage of our method over RNN even for 1d periodic problems. These results will be added to the final version.

**Stopping Criterion in Experiments** (R4): Our stopping criterion is chosen fairly and reasonably: all experiments are stopped at the time when the training loss of all the methods stop to decrease and becomes a constant, i.e., when the model has converged. The performances of this converged model is not visibly different from the early stopping point for the experiments we considered. For example, the goal of Figure 7 (for the atmospheric experiment) and Figure 14.b (EUR-USD experiment in the appendix) are plotted to order to show that we stopped at the point when the model converged, moreover, neither of our model or the baselines seem to suffer significantly from overfitting, judging from these two figures. Therefore, the comparison is indeed fair and reasonable. In our final version, we will add similar plots for other experiments to clarify the stopping criterion for the experiments.

**Comparison to Swish and Leaky-ReLU Seems Necessary** (R5): The proposed method indeed outperforms Swish and Leaky-ReLU significantly for tasks in section 6.2 and 6.3. This is because Swish and Leaky-ReLU suffer from the same problem as ReLU, which is guaranteed by our theorem and by the discussion above. Therefore, we did not include them for visual clarity. Their performance on the tasks is now shown in Figure 2. We see that, for Swish and Leaky-ReLU, the learning is hard to mismatched inductive bias, and this leads to their inferior performance. We will add this plot to the appendix in the final version to avoid confusion.

**Strength of the claim** (R5): 1. **Applicability of the theorem**. We agree that some qualifier is needed in this claim. On the one hand, the specific statement of the theorem applies to $\tanh$ and ReLU, but it is general as discussed in line 64-66. Since the proof is only based on the asymptotic property of activation function when $x \to \pm\infty$, one can prove the same theorem for any continuous activation function that asymptotically converges to a $\tanh$ or ReLU; for example, this would include Swish and Leaky-ReLU (and almost all the other ReLU-based variants), which converge to ReLU; one can follow exactly the same proving procedure to prove a similar theorem for each of these activation functions. On the other hand, we agree that this statement might be too strong, and we will state the above condition clearly for the applicability of our theory in the final version to avoid confusion. 2. **Generality of Snake.** Here, we say that Snake is more "general" in the sense that it is not only capable of approximating a function in the bounded region, but also capable of extrapolating beyond a bounded region (in a periodic way). This claim does need some qualification to clarify, and we will modify this claim in the final version. **Some details. Some Problems in the Appendix** (R1): We will add more details and correct grammatical errors to the Appendix sections.

(a) Training set, with $\sigma^2 = 0.2$, $T = 200$.

(b) Performance vs $\sigma^2$.

(c) Single update computation time vs. length of the data point.

Figure 1: Comparison of RNN with feedforward neural network with Snake.

(a) Same as Fig. 6 in the paper. (b) Training loss and testing loss.

Figure 2: Comparison with Swish etc..

[Meta-Review · NeurIPS 2020]

I think this is an interesting submission, that lead to a detailed discussion among the reviewers. Overall the work is novel and looks at an interesting question regarding extrapolation (and dealing with periodic functions). Overall I agree with some of the reviewers that the motivation and generally the write-up of the work could be improved, but I think there is already value in the work. I would like to highlight a few points that are worth considering: * a further discussion regarding how the proposed approach compares to RNNs or autoregressive models when it comes to modeling periodicity * there are some concerns regarding the methodology used (e.g. stopping criterion used) that would be nice to be better clarified in the final version of the work In general, please try to incorporate as much as you can from the clarifications in the rebuttal, and try to answer as many of the worries brought up by the reviewers in order for the paper to have the impact it deserves.